# Upregulation of Sulfated N-Glycans in Serum as Predictive Biomarkers for Early-Stage Breast Cancer

**DOI:** 10.3390/ijms26114968

**Published:** 2025-05-22

**Authors:** Dereje G. Feleke, Bryan M. Montalban, Solomon T. Gizaw, Hiroshi Hinou

**Affiliations:** 1Laboratory of Advanced Chemical Biology, Graduate School of Life Science, Hokkaido University, Sapporo 001-0021, Japan; derejegetachew.feleke.g0@elms.hokudai.ac.jp; 2Department of Physical Sciences and Mathematics, College of Arts and Sciences, University of the Philippines Manila, Padre Faura St., Manila 1000, Philippines; bmmontalban@up.edu.ph; 3Department of Biochemistry, College of Health Science, Addis Ababa University, Addis Ababa P.O. Box 9086, Ethiopia; solomon.tebeje@aau.edu.et; 4Frontier Research Center for Advanced Material and Life Science, Faculty of Advanced Life Science, Hokkaido University, Sapporo 001-0021, Japan

**Keywords:** breast cancer, sulfated N-glycans, glycoblotting, MALDI-TOF MS, biomarkers, early detection

## Abstract

Breast cancer (BC) is a major global health concern, and early detection is key to improving patient outcomes. Aberrant glycosylation, particularly the sulfation of glycans, is implicated in cancer progression; however, analyzing these low-abundance glycans is challenging. This study aimed to profile serum sulfated N-glycans in Ethiopian patients with BC to identify novel biomarkers for the early detection of BC. Using a glycoblotting-based sulphoglycomics workflow, including high-throughput glycoblotting enrichment, weak anion exchange (WAX) separation, and MALDI-TOF MS, serum samples from 76 BC patients and 20 healthy controls were analyzed. Statistical evaluation revealed significant differences in the sulfated N-glycan profiles. Seven mono-sulfated N-glycans were markedly elevated in patients with BC, demonstrating high diagnostic accuracy (AUC ≥ 0.8) in this internal cohort. Terminal Lewis-type glycan epitopes were prominent in sulfated glycans but were absent in their non-sulfated counterparts. The increased fucosylation and sialylation of sulfated glycans are statistically significant markers of early-stage BC. The preservation of sialic acid groups during the analysis ensured detailed structural insight. This pioneering study quantitatively examined sulfated N-glycans in BC and identified potential glyco-biomarkers for early detection. Validation in larger, diverse cohorts is needed to establish their broader diagnostic relevance and improve our understanding of cancer-associated glycomic alterations.

## 1. Introduction

Breast cancer (BC) is the most common cancer among women and a leading cause of cancer-related mortality worldwide [1]. Early detection and diagnosis are crucial for improving the survival outcomes of patients with BC. Currently, imaging techniques such as mammography, magnetic resonance imaging (MRI), and ultrasonography are the mainstays of BC screening [2,3]. However, these methods may not always facilitate an early diagnosis. In addition to imaging, serum biomarkers such as cancer antigen 15-3 (CA 15-3) and carcinoembryonic antigen (CEA) have been employed for BC detection. Unfortunately, these biomarkers have limited sensitivity and specificity and are primarily used to monitor treatment responses in patients with metastatic BC [4,5]. Therefore, there is an urgent need for highly sensitive and specific biomarkers to aid in the early detection of BC.

Glycosylation, a common post-translational modification (PTM), plays a crucial role in various biological processes, including molecular recognition, cell adhesion, migration, immune regulation, and receptor signaling [6,7,8]. Altered glycosylation patterns, frequently observed in cancer, contribute to tumor cell proliferation, differentiation, invasion, and metastasis [9,10,11]. These changes often correlate with tumor burden and poor prognosis, making glycosylation analysis a valuable tool for distinguishing normal from malignant cells [2,7]. Our group previously investigated changes in serum non-sulfated glycan profiles associated with the invasive and metastatic potential of BC [12].

Beyond core glycan biosynthesis pathways, additional modifications such as sulfation, phosphorylation, and acetylation can occur, with sulfation being one of the most common post-glycosylation modifications [13]. Sulfated glycans are involved in various physiological processes, including lymphocyte homing [14] and hormone clearance [15], as well as in diseases such as cystic fibrosis [16], osteoarthritis [17], and cancer [18,19]. Recent studies have suggested that trace levels of sulfated N-glycans from immunoglobulin G (IgG) may serve as potential biomarkers of rheumatoid arthritis [20]. Zhang et al. demonstrated that cell surface 6-sulfosialyl LewisX is closely associated with [21] metastasis, suggesting its potential as a novel marker of BC progression [21]. These findings underscore the importance of sulfated glycans as promising biomarker candidates and highlight the need for further analysis to discover novel BC biomarkers and to better understand disease mechanisms [22,23].

Despite their critical physiological roles, sulfated glycans are challenging to analyze because of their low abundance in mammalian cells, tissues, and organisms [24]. Although mass spectrometry (MS)-based methods are widely regarded as robust tools for glycomic profiling, the detection and analysis of sulfated glycans present specific challenges. The ion suppression effects caused by neutral and sialylated non-sulfated glycans in both positive- [23] and negative-ion modes, as well as [24,25] the labile nature of sulfate groups, complicate their ionization and detection [26]. To overcome these limitations, techniques such as permethylation followed by anion exchange separation have been developed [23,26], allowing for the selective detection of sulfated glycans in the negative mode [23,25,26,27]. However, these methods involve time-consuming sample handling and multiple cleanup steps, resulting in sample loss [24]. Alternative methods, such as serotonin-immobilized columns, have shown promise for analyzing labeled minor acidic glycans. However, these techniques have drawbacks, particularly their inability to provide information about the sialylation status of glycans owing to prior neuraminidase digestion [28].

In this study, we utilized a glycoblotting-based sulphoglycomics workflow to address these limitations. This method enables a comprehensive analysis of sulfated and phosphorylated glycans in biological samples. It is particularly effective in enriching benzyloxyamine (BOA)-labeled minor acidic glycans while simultaneously separating sulfated and phosphorylated glycans [29,30]. Conventional methods often struggle to distinguish between sulfated and phosphorylated glycans because of their similar physicochemical properties and molecular weights (H_2_PO_3_ = 79.9663 Da and HSO_3_ = 79.9568 Da) [28,31]. The glycoblotting approach circumvents this issue by providing an efficient and robust platform for the detection of challenging glycan species.

This study represents the first application of the glycoblotting-based sulphoglycomics workflow to examine negatively charged sulfated N-glycans in the serum glycoproteins of patients with BC. By comparing the sulfated glycan profiles of healthy individuals and patients with BC, we identified sulfated N-glycan species associated with BC. These findings have significant diagnostic and prognostic implications and offer potential utility in early BC detection and patient management.

## 2. Results

### 2.1. Identification of Sulfated N-Glycans Using a Glycoblotting-Based Sulphoglycomics Workflow

To comprehensively profile sulfated N-glycans in human serum, we employed our previously established glycoblotting-based sulphoglycomics workflow (Appendix A) [30]. This approach enabled the selective enrichment and precise structural analysis of sulfated N-glycans, overcoming the challenges posed by their low abundance. Serum N-glycans were isolated using a glycoblotting-based workflow. Glycoproteins were enzymatically digested with PNGase F to release N-glycans, which were captured on hydrazide-functionalized beads. To minimize interference from non-sulfated glycans, particularly sialylated species, during weak anion exchange (WAX) separation and MALDI-TOF MS analysis, 3-methyl-1-p-tolyltriazene (MTT) was used to methyl-esterify the carboxyl groups of sialic acids [27,32,33]. This modification selectively neutralizes sialic acids while preserving the negative charge of sulfated glycans, enabling efficient separation. The differential charge properties facilitate the enrichment of sulfated glycans, which are otherwise difficult to isolate from more abundant sialylated species in negative-mode MALDI-TOF MS

Prior to mass spectrometric analysis, neutral and sialylated N-glycans were eluted using 1% acetic acid in 50% acetonitrile (ACN). Sulfated N-glycans were subsequently eluted with 1% ammonium hydroxide in 5% ACN (pH 10.5), allowing selective detection and structural characterization. This optimized approach provides a robust and reproducible platform for the high-sensitivity analysis of sulfated N-glycans in complex biological samples.

In the initial analysis, we performed MALDI-TOF MS in the negative-ion mode to analyze sulfated N-glycans from the serum of healthy individuals using the glycoblotting-based sulphoglycomics workflow. Prior to enrichment, only non-sulfated N-glycans, including their sialylated forms, were detected, whereas sulfated N-glycans remained undetectable in the mass spectrum. After enrichment, several distinct sulfated N-glycans were identified as [M − H]^−^ ions, demonstrating the sensitivity of MALDI-MS for sulfated glycans (Figure 1, Appendix A). This result underscores the challenge of detecting sulfated glycans in serum due to their suppression by more abundant non-sulfated glycans.

Among the identified sulfated N-glycans, several mono-sulfated and di-sulfated structures were observed, with the latter likely contributing to enhanced interactions with glycan-binding proteins, which are known to facilitate cancer cell migration and invasion [34]. Interestingly, several neutral N-glycan peaks observed in the unenriched fraction (*m*/*z* 1752.609, 2219.717, 2378.959, and 2524.822) exhibited corresponding sulfated counterparts in the enriched acidic glycan fraction, each showing a characteristic mass difference of approximately 56 Da [SO_3_–Na^+^–H^+^]. This mass shift provides strong evidence for sulfate modifications on these glycans. Phosphorylated N-glycans were not observed. Most sulfated N-glycans are complex-type structures, characterized by non-reducing terminal sulfo-LacNAc, sulfosialyl-LacNAc, sulfo-Lewis-type, and sulfosialyl-Lewis-type glycan epitopes. Non-sulfated N-glycans typically lack Lewis-type epitopes, and monofucosylated structures are often difficult to distinguish from core fucosylation. In contrast, major sulfated N-glycans frequently carry multiple fucose residues, forming terminal Lewis-type epitopes. Sulfated N-glycans without Lewis epitopes also exhibited distinct signal intensity patterns, independent of non-sulfated glycans. These findings highlight the potential of the glycoblotting-based sulphoglycomics workflow to uncover novel sulfated glycan biomarkers with important clinical applications in early BC detection. The selective identification of sulfated glycans with Lewis-type and other terminal epitopes suggests that these structures may offer improved diagnostic accuracy, particularly in early-stage BC.

### 2.2. MALDI-TOF/TOF Profiling of Sulfated N-Glycans in Human Serum

In the positive-ion reflectron mode, sulfated N-glycans were analyzed using MALDI-TOF/TOF MS, where they were detected as singly charged disodiated molecular ions, [M + 2Na − H]^+^. The TOF/TOF spectra provided comprehensive structural information through glycosidic bond and cross-ring cleavages, facilitating precise sequence assignments. Diagnostic fragment ions and neutral losses confirmed the structures of the BOA-labeled sulfated N-glycans. Figure 2 presents the MALDI-TOF/TOF MS profile, highlighting the mono-sulfated complex-type N-glycans detected in human serum samples.

For the mono-sulfated N-glycan with a precursor ion at *m*/*z* 1855, a characteristic neutral loss of 472 Da suggested the elimination of a BOA-labeled reducing-end GlcNAc-bearing core fucosylation. This was followed by the sequential loss of GlcNAc (*m*/*z* 203) from the core structure. Fragment ions retaining sulfate modifications were detected, with the loss of *m*/*z* 328 corresponding to the elimination of mono-sulfated GlcNAc [SO_3_ + GlcNAc + 2Na]^+^ at the non-reducing terminus. Additional fragmentation revealed the loss of *m*/*z* 490 and *m*/*z* 694, indicating the successive removal of hexose and HexNAc residues from the non-reducing end. Core mannose residues were sequentially cleaved (with a loss of 486 *m*/*z*), whereas molecular ion peaks at *m*/*z* 388, 916, and 1078 represented the characteristic loss of sodium sulfite (*m*/*z* 102) from the precursor ions at *m*/*z* 490, 1018, and 1180. This fragmentation pattern was consistent with the presence of sulfate groups in the analyzed structure. Furthermore, cross-ring cleavage at the reducing terminus of BOA-labeled GlcNAc resulted in the loss of *m*/*z* 206 (Figure 2).

Similarly, the TOF/TOF spectra of the mono-sulfated complex-type N-glycan at *m*/*z* 2626.9 provided informative sequence assignments for the corresponding peptide. Similar to the *m*/*z* 1855 fragmentation pattern, a neutral loss of *m*/*z* 472 and subsequent elimination of the core GlcNAc residues were observed. Sulfate-modified diagnostic fragment ions were identified, including the loss of *m*/*z* 327, which represents sulfated GlcNAc at the non-reducing terminal. A neutral loss of *m*/*z* 305 indicates the removal of methyl-esterified sialic acid. Further cleavage of the mannose residues was evident in the fragment ions at *m*/*z* 958, 1120, and 1282, corresponding to [SO_3_ + Hex + GlcNAc + Hex + Me−NeuAc + 2Na]^+^, [SO_3_ + Hex + GlcNAc + 2Hex + Me−NeuAc + 2Na]^+^, and [SO_3_ + Hex + GlcNAc + 3Hex + Me−NeuAc + 2Na]^+^, respectively. Additionally, the molecular ion peaks at *m*/*z* 388, 694, and 1018 reflect the characteristic sodium sulfite losses (*m*/*z* 102) from the precursor ions at *m*/*z* 490, 796, and 1120.

A similar fragmentation pattern was observed for the sulfated N-glycan at *m*/*z* 2217, where sulfosialyl-LacNAc was detected at the non-reducing end. In contrast, the sequential loss of two HexNAc residues on the opposite arm suggests the presence of LacdiNAc or triantennary N-glycans. The neutral loss of *m*/*z* 326 confirmed the elimination of the reducing-end GlcNAc residue labeled with BOA. These consistent fragmentation patterns provide compelling evidence of sulfate modifications in the analyzed glycan structures, reinforcing their potential as biomarkers of BC.

### 2.3. Diagnostic Potential of Identified Sulfated N-Glycans: Quantitative Analysis of Serum Sulfated N-Glycans in Healthy Control Individuals and BC Patients

To assess the potential of sulfated N-glycans as biomarkers for BC, we conducted a comparative analysis of serum samples from 76 patients with BC and 20 age-matched normal controls (NCs). The clinical characteristics of the study participants are summarized in Appendix A. Representative MALDI-TOF MS profiles of BOA-labeled sulfated N-glycans from both BC and NC serum samples are shown in Figure 3. These glycans were enriched using our glycoblotting-based workflow and fractionated via WAX separation to enhance detection sensitivity.

Eight mono-sulfated and five di-sulfated N-glycans were consistently detected across all study samples (Appendix A). To ensure analytical reliability, the quantitative reproducibility of each glycan peak was assessed using serially diluted standard serum samples (0.2×, 0.4×, 0.6×, 0.8×, 1×, and 1.2×). The peak area of each glycan was normalized to a fixed concentration of the internal standard, and standard calibration curves were generated (Appendix A). Only glycan peaks that exhibited linearity and minimal outlier deviations were selected for further statistical analysis.

To quantitatively assess sulfated N-glycan expression in patients with BC and NCs, we employed an internal standard, a mono-sulfated desialylated biantennary complex-type N-glycan (129 pmol), synthesized from sialoglycopeptide (SGP) via chemoenzymatic methods [35]. A complete list of the identified sulfated N-glycans, including their composition and structure, is provided in Appendix A.

Our analysis revealed that sulfated N-glycans in human serum were predominantly complex-type, with hybrid-type structures accounting for approximately 1%. Among the complex N-glycans, biantennary glycans were the most abundant. Notably, ≥75% of the sulfated glycans were fucosylated and ≥66% were sialylated, underscoring the structural complexity of these glycoforms.

Comparative quantification revealed that all 13 sulfated N-glycans were upregulated in BC sera compared to in NC samples (Figure 4). The expression levels were determined by normalizing the peak area of each glycan to the fixed concentration of an internal standard. Subsequent statistical analysis (independent *t*-test) demonstrated that seven mono-sulfated and two di-sulfated N-glycans were significantly elevated in patients with BC (Table 1).

### 2.4. Evaluation of the Diagnostic Performance of Serum Sulfated N-Glycans

To assess the diagnostic potential of serum sulfated N-glycans, we performed receiver operating characteristic (ROC) curve analysis on glycans that exhibited statistically significant differences (*p* ≤ 0.05) in expression based on t-test analysis. The area under the curve (AUC) was calculated to evaluate the ability to distinguish BC patients from NCs. Sulfated N-glycans that met both the *p* ≤ 0.05 and AUC ≥ 0.8 criteria were considered potential biomarkers. The seven mono-sulfated N-glycans demonstrated excellent diagnostic accuracy and were identified as potential biomarkers (Table 2). Notably, these glycans exhibited a strong discriminatory power, particularly in early-stage BC (stages I and II). Specifically, four mono-sulfated LacNAc-terminated N-glycans (*m*/*z* 1809, 2077, 2272, and 2450) and three mono-/bi-sialylated N-glycans (*m*/*z* 2171, 2276, and 2581) effectively distinguished stage I BC patients from NCs, while three sulfated glycans (*m*/*z* 1809, 2077, and 2272) showed significant differences in stage II patients. Interestingly, no significant biomarker candidates were detected in patients with stage III BC, whereas four sulfated N-glycans (*m*/*z* 2077, 2171, 2276, and 2581) were identified as potential biomarkers in patients with stage IV BC (Figure 5). Additionally, three sulfated N-glycans (*m*/*z* 2077, 2276, and 2581) consistently differentiated BC patients from NCs at all stages. Multiple comparisons were conducted to visualize the sulfated N-glycan abundance across different BC stages, revealing the most significant increases in stages I and II. As shown in Figure 6, the dot plots and ROC curves further highlight the strong diagnostic potential of these biomarkers in early-stage BC. These findings underscore the clinical relevance of sulfated N-glycans as promising candidates for early detection and disease stratification in BC.

### 2.5. Serum Glyco-Subclass Patterns as Predictive Indicators for Early-Stage BC

In addition to the analysis of individual sulfated N-glycans, we examined the glycan subclass patterns to assess the structural similarities and overall trends in BC patients compared with NCs. Our findings revealed significant variations in glycan subtypes—including fucosylated; mono-, bi-, and overall sialylated; and overall mono-sulfated N-glycans—particularly in patients with stage I and II BC (Figure 7). Notably, ROC analysis demonstrated that several glycan subclasses are strong diagnostic indicators of early-stage BC. In stage I patients, fucosylated; mono-, bi-, and overall sialylated; and overall sulfated N-glycans exhibited high AUC values—0.979, 0.944, 0.870, 0.891, and 0.941, respectively (*p* < 0.0001)—indicating excellent discriminative ability. Similarly, in stage II patients, fucosylated glycan subclasses showed a high diagnostic potential, with an AUC value of 0.945. These findings align with the expression patterns observed for individual sulfated N-glycan biomarkers, reinforcing the significance of early-stage glycomic alterations in BC. The increased abundance of these specific glycotypes suggests their potential roles in BC progression and early detection.

## 3. Discussion

An increasing body of evidence highlights the critical role of sulfation in modifying the non-reducing terminal epitopes of N-glycans. These sulfate modifications significantly alter the physicochemical properties of glycans, influencing their capacity to act as specific binding sites for pathogens and endogenous glycan-binding proteins [27,36]. Changes in the structure or abundance of sulfated glycans can initiate profound physiological and pathological processes [20]. In the context of cancer, terminal sulfate modifications are associated with malignancy and play crucial roles in tumor progression and metastasis [34,37]. Given these insights, the analysis of sulfated N-glycans holds substantial promise for uncovering new biomarkers and deepening our understanding of the mechanisms underlying BC.

Although our study focused on an Ethiopian cohort, it establishes a robust foundation for future large-scale investigations—incorporating cross-validation, external replication, and correction for false discovery rates—across diverse populations and clinical settings, including the use of benign breast conditions as a control. These insights not only advance our understanding of cancer glycobiology but also reinforce the potential of sulfated glycan profiling as a powerful tool for BC diagnosis. Further research is warranted to validate these biomarkers in broader populations and to elucidate their mechanistic roles in BC pathogenesis.

In this study, we employed a glycoblotting-based sulfoglycomics workflow to examine the serum sulfated N-glycans of Ethiopian patients with BC and matched healthy controls. This approach enabled us to successfully identify and quantify several previously unreported mono- and di-sulfated N-glycans in the serum of patients with BC. The high glycan-capturing efficiency of the glycoblotting method coupled with the robust sulfated N-glycan enrichment provided by weak anion exchange (WAX) separation was instrumental in achieving these findings. We combined this workflow with a synthetic internal standard to achieve an efficient quantitative analysis of sulfated N-glycans rich in sialic acid and Lewis-type antigens in serum. This study offers new insights into the glycomic landscape in BC, highlighting the potential significance of sulfated glycans.

Our comparative analysis revealed that seven mono-sulfated and two di-sulfated N-glycans were significantly upregulated in the sera of BC patients compared with healthy controls. ROC analysis demonstrated that the seven mono-sulfated N-glycans exhibited excellent diagnostic performance in distinguishing patients with early-stage BC from healthy controls (Table 2, Figure 5). These glycans included mono-sulfated LacNAc (*m*/*z* 1809, 2077, 2272, and 2450) and both mono- and di-sialylated termini (*m*/*z* 2171, 2276, and 2581). The ability of these glycans to accurately distinguish early-stage BC is particularly noteworthy, as early detection is key to improving patient outcomes through timely interventions and treatments. Furthermore, based on structural similarity, serum glycotyping patterns exhibited excellent diagnostic performance for predicting early-stage BC in specific glycan subtypes. Notably, for stage I, the glycotyping patterns—such as fucosylated; mono-, bi-, and overall sialylated; and overall sulfated—demonstrated strong predictive capabilities. In Stage II, fucosylated glyco-subclasses were strong predictors. These findings suggest significant alterations in the biosynthetic pathways of sulfated glycans in patients with BC, which begin at an early stage of the disease. Similarly, an increase in sulfated glycans was also noted in patients with pancreatic cancer [28], suggesting that these glycomic alterations may be a common feature in the development and progression of various cancers. Identifying the molecular pathways involved in these changes could provide valuable insights into BC biology and reveal novel therapeutic targets.

Furthermore, the significance of our findings was underscored by the discovery of two mono-sulfated N-glycans (*m*/*z* 2276 and 2581) in human IgG serum samples, as reported by Wang et al. [20]. Moreover, similar to our results, the mono-sialylated complex-type N-glycan (*m*/*z* 2276) has been recognized as a potential biomarker for rheumatoid arthritis. This was further supported by the fact that these two sulfated glycans and *m*/*z* 1809 were previously reported by our group in the neutral (non-sulfated) forms of human IgG within the same samples [12]. This indicates that the three identified biantennary sulfated N-glycans may have originated from the serum IgG. These findings strongly suggest an intricate connection between the immune system and alterations in serum sulfate N-glycosylation patterns during BC progression. To further validate our findings, we quantified the sulfated N-glycans exclusively derived from IgG in the BC serum samples. Several sulfated N-glycans associated with IgG were detected, confirming that these glycan modifications are intimately related to immune responses and are implicated in BC development (Appendix A).

Given that IgG is a major serum protein and an essential component of the humoral immune system, N-glycans with sulfation modifications play a crucial role in immune recognition [38] and can significantly impact the function of IgG, potentially by altering the structure of the Cγ2 domain [20]. However, further studies are needed to investigate the effect of sulfated N-glycans on IgG and their association with BC progression. In addition to IgG, carriers of other candidate sulfated biomarkers in human serum have yet to be identified.

We used the same serum samples previously examined by Gebrehiwot et al. in their study of Ethiopian patients with BC [12]. Although their analysis focused on identifying non-sulfated N-glycans as potential biomarkers for early BC detection, our investigation revealed a distinct set of sulfated N-glycans. Our findings included specific sulfated glyco-epitopes that were not identified in their study, indicating a complementary and enriched perspective of the glycomic landscape associated with BC. Their analysis identified seventeen candidate biomarkers, whereas our study identified seven sulfated N-glycans that met the criteria for biomarker identification. This suggests a low abundance of sulfated N-glycans. However, despite their lower abundance, sulfated N-glycans exhibited excellent diagnostic accuracy for the early detection of BC compared to that of the control groups (Table 2). This insight could provide an additional dimension for identifying potential biomarkers for the early detection of BC.

Compared to their non-sulfated counterparts, carbohydrate sulfation has demonstrated increased potency as a ligand for glycan-binding proteins such as selectins [13] and galectins [39]. Studies have illustrated the enhanced binding affinity of siglecs to their sulfated glycan ligands in various cancer cells, including BC cells, potentially enhancing their immune invasion capabilities [34]. It has also been shown that the synthesis of 6-sulfosialyl LeX on the cell surface is implicated in facilitating BC migration and invasions [21]. These findings strongly suggest that the enhanced sulfation of terminal epitopes on N-glycans promotes BC progression and metastasis. Similarly, our results showed that sulfated glycans predominantly displayed terminal sulfo-LacNAc, sulfosialyl-LacNAc, sulfo-Lewis-type, and sulfosialyl-Lewis-type glycan epitopes on N-glycans, which were upregulated in BC. However, the presence of Lewis-type antigen terminal structures in non-sulfated N-glycans in the serum was negligible. These modifications may have important implications for the aggressiveness and invasiveness of BC. The upregulation of fucosyltransferases leads to the increased expression of terminal glycan epitopes on transformed cells, which can bind to macrophages and immature dendritic cells through the C-type lectin DC-SIGN. This interaction significantly affects the function of these immune cells. Consequently, targeting these glycans on tumor cell glycoproteins is a promising strategy for enhancing antitumor immune responses [40]. Furthermore, the increased expression of these glycan epitopes can influence various aspects of cancer cell biology, including the epithelial–mesenchymal transition, immune interactions, the induction of multidrug resistance, and cancer stemness [41]. However, further investigation is necessary to fully elucidate the specific mechanisms and downstream effects of sulfated N-glycans on BC progression and invasion. Focusing on these N-glycan pathways may offer a promising therapeutic approach for BC [4]. Targeting the sulfotransferase enzyme responsible for synthesizing specific sulfated N-glycans may impede BC progression and metastasis. Moreover, the analysis of these glycans did not result in the conclusive structural identification of sulfated glycans. This difficulty is particularly pronounced in samples with a low content of sulfated N-glycans, where the structural information obtained is markedly limited. In the future, we will explore advanced techniques, such as lectin affinity chromatography and more rigorous validation (e.g., exoglycosidase digestion and LC-MS/MS), to enhance the isolation and structural elucidation of these intricate glycan modifications. While this study focused on BC, the glycoblotting-based sulfoglycomics approach has potential utility in profiling sulfated N-glycans across various cancer types, where glycosylation alterations also play critical roles. Although clinical pathology data such as tumor grade and receptor status (ER, PR, HER2) were not available at the time of this study, we aim to incorporate this information in future analyses to enable deeper clinicopathological associations. To further evaluate potential confounding factors, age-stratified analysis within the BC cohort revealed that three biomarker glycans (*m*/*z* 2272, 2276, and 2581) showed significant age-associated variation but remained elevated in early-stage BC. These findings (Appendix A) support the need for age-adjusted validation in future studies. Additionally, we are currently analyzing paired tumor and adjacent tissues to explore the correlations between serum and tissue sulfated N-glycan profiles and their associations with histopathological features.

## 4. Materials and Methods

### 4.1. Study Population and Sample Collection

Human serum samples were collected from female BC patients (n = 76) across stages I-IV (stage I n = 17, stage II n = 20, stage III n = 17, stage IV n = 22) and healthy controls (n = 20) in Ethiopia during 2015–2016. Healthy, age-matched individuals free from any known disease were included in the control group. Informed consent was obtained from all participants, and this study was conducted in accordance with the ethical standards of the Declaration of Helsinki. Ethical approval was granted by the Review Boards of Addis Ababa University, School of Medicine (Ethiopia), and Hokkaido University, Faculty of Advanced Life Sciences (Japan). To maintain sample integrity, the serum samples were stored at −80 °C in plastic vials, transported to Japan within 72 h, and packed with dry ice in a foam box.

### 4.2. Materials

Peptide N-glycosidase F (PNGase F) was purchased from New England BioLabs (Ipswich, MA, USA). Ammonium bicarbonate (ABC), 3-methyl-1-*p*-tolyltriazene (MTT), 2,5-hydroxybenzoic acid (DHB), benzyloxyamine hydrochloride (BOA-HCl), sodium bicarbonate (NaHCO_3_), trifluoroacetic acid (TFA), and 3,4-diamino benzophenone (DABP) were purchased from Tokyo Chemical Industry Co. (Tokyo, Japan). BlotGlyco^®^ H beads were obtained from Sumitomo Bakelite Co., Ltd. (Tokyo, Japan). Proteinase K was purchased from Roche (Mannheim, Germany), and trypsin was sourced from Sigma-Aldrich Corp. (St. Louis, MO, USA). MultiScreen Solvinert filter plates were purchased from Millipore Co. (Billerica, MA, USA).

### 4.3. N-Glycan Release from Human Serum Glycoprotein [30,42]

Whole serum from each sample was subjected to enzymatic pretreatment to generate glycans with reducing terminals. This enzymatic pretreatment facilitated subsequent chemical ligation with hydrazide-functionalized BlotGlyco^®^ H beads during glycoblotting. Hence, 15 µL of human serum and 10 µL of a 12.9 µM mono-sulfated desialylated N-glycan prepared from SGP (Tokyo Chemical Industry Co., Ltd.), used as an internal standard, were mixed with 50 µL of 200 mM ammonium bicarbonate (NH_4_HCO_3_). Four microliters of denaturing buffer (5% sodium dodecyl sulfate, 0.4 M dithiothreitol) was added to denature the sample, followed by heating at 100 °C for 10 min. Subsequently, 10 µL of 123 mM iodoacetamide (IAA) was added, and the mixture was incubated at room temperature in the dark for 1 h. The mixture was then digested by adding 10 µL trypsin (40 U/µL) dissolved in 1 mM HCl and incubated overnight at 37 °C. Subsequently, the enzyme activity was halted by heating at 90 °C for 10 min and allowing to cool to room temperature. To initiate further processing, 8 µL of reaction buffer (0.5 M Na_3_PO_4_, pH 7.5) and 10% NP-40 were added sequentially and incubated for 10 min at 37 °C. N-glycans were released using 2 µL of PNGase F with an activity of 5 U/µL, followed by overnight incubation at 37 °C. For further digestion, 10 µL of proteinase K (0.5 U/µL) was added, and the mixture was incubated for 3 h at 37 °C. The enzyme was then heat-inactivated at 90 °C for 10 min. The resulting sample was dried using SpeedVac (EYELA, Tokyo, Japan) and stored at −20 °C until further use.

### 4.4. Enrichment of Glycans Using Glycoblotting [30,42,43,44]

A 250 µL suspension of BlotGlyco^®^ H beads at a concentration of 10 mg/mL in water was aliquoted into 96 wells of a MultiScreen Solvinert filter plate, and the water was removed by applying a vacuum. The dried sample containing released N-glycans was reconstituted by adding 20 µL of Milli-Q water. The reconstituted sample (20 µL) was added to each well, followed by 180 µL of 2% acetic acid in acetonitrile (AcOH/ACN). The plates were then incubated at 80 °C for 45 min. Following incubation, two successive washing steps were performed using 200 µL each of 2 M guanidine HCl in 16.6 mM NH_4_HCO_3_, water, and a 1% solution of triethylamine in methanol. The unreacted hydrazide functional groups were acetyl-capped by adding 100 µL of 10% acetic anhydride in methanol at room temperature for 30 min. Any residual acetic anhydride was removed under vacuum. Each well was washed twice with 10 mM hydrochloric acid (HCl), methanol, and dioxane. For on-bead methyl esterification, 100 µL of 100 mM MTT in dioxane was added, and the mixture was incubated at 60 °C for 90 min. Subsequently, two washing steps were performed using 200 µL of dioxane, water, methanol, and water. To facilitate effective labeling through a trans-iminization reaction, 20 µL of 50 mM BOA-HCl and 180 µL of 2% AcOH/ACN solution were added sequentially. The mixture was then incubated at 80 °C for 45 min. N-glycans labeled with BOA were eluted twice with water (150 µL). The resulting glycan solution was dried using SpeedVac and stored at −20 °C until further use.

### 4.5. Enrichment of Sulfated N-Glycans Using WAX [29,30]

Sulfated N-glycan enrichment was performed as previously described [30]. A small cotton plug was inserted into a 200 µL micropipette tip. Subsequently, 50 µL of 3-aminopropyl silica gel (Tokyo Chemical Industry Co., Ltd., 100 mg/mL suspension) was loaded onto a micropipette tip, on top of a cotton plug. The gel was allowed to settle under gravity. The packed WAX microcolumn was conditioned and washed twice with 100 µL of water, ACN, and 1% AcOH in 95% ACN. After each washing step, the mixture was centrifuged at 500 rpm for 2 min. To reconstitute the BOA-labeled N-glycans obtained from glycoblotting, 20 µL of Milli-Q water was used. A 5 µL sample aliquot was then dissolved in 150 µL of 1% AcOH in 95% ACN and loaded onto the column. The sample was allowed to elute under gravity. The collected eluate was reloaded onto the column, and this process was repeated three times to ensure adequate interaction. The column was then washed with 150 µL of 1% AcOH in 95% ACN, followed by centrifugation at 500 rpm for 2 min to remove unbound and hydrophobic contaminants. Then, 150 µL of 1% AcOH in 50% ACN was added to elute BOA-labeled neutral and methylated sialylated N-glycans. The elution step was performed twice, and after each elution, the samples were centrifuged at 500 rpm for 2 min. Sulfated N-glycans were eluted using 150 µL of 1% NH_4_OH in 5% ACN (pH 10.5). As in the previous steps, this elution was repeated twice, and centrifugation was performed at 500 rpm for 2 min. Finally, the resulting glycan solution was dried using SpeedVac and stored at −20 °C until use.

### 4.6. MALDI-TOF MS Analysis

The mass spectrometric analysis of BOA-labeled sulfated N-glycans was performed using an Ultraflex III (Bruker, Bremen, Germany) instrument equipped with a reflector in the negative-ion mode. DABP (10 mg/mL) in 75% ACN with 0.1% TFA was used as a matrix throughout this work [25,45], from which 0.5 µL of each sample and matrix solution was spotted onto an MTP 384 target plate (polished steel TF, Bruker) and dried at room temperature. MALDI-TOF/TOF MS was performed in positive mode using 10 mg/mL DHB/NaHCO_3_ (10:1) in a 50% ACN matrix. To ensure reproducibility, each sample was spotted in four replicates, and each spectrum was generated by accumulating 1000 laser shots per sample. The average of the four normalized data points for each sample was used for statistical analysis. Data acquisition and processing were performed using the Flexanalysis software (version 3.0; Bruker, Bremen, Germany, S/N = 2). The intensity of the monoisotopic peak for each sulfated N-glycan was normalized to 12.9 µM of an internal standard (mono-sulfated biantennary complex-type N-glycans synthesized using chemoenzymatic methods [35]). The normalized data were used for further statistical analysis and quantitative comparison of sulfated N-glycan profiles. The structural compositions of the sulfated N-glycans were assigned using the Expasy GlycoMod Tool and Glyconnect Database provided by the Swiss Institute of Bioinformatics (https://web.expasy.org/glycomod/ (accessed on 4 September 2023) and GlycoWorkbench [46,47]. GraphPad Prism version 5 software (GraphPad Software, San Diego, CA, USA) was used for data analysis and graph plotting.

### 4.7. Statistical Analysis

A comparison between the two groups (NC vs. entire BC group) was performed using an independent-sample t-test. Multiple comparisons between the NC and clinical stage groups were conducted using Bonferroni’s one-way analysis of variance (ANOVA). To evaluate the diagnostic potential of individual glycans or glyco-subclasses that showed significant differences, a receiver operating characteristic (ROC) test was performed. The area under the curve (AUC) generated from the ROC test was used to measure the diagnostic accuracy of potential glycan biomarkers. AUC values within the ranges of 0.9–1, 0.8–0.9, 0.7–0.8, and <0.7 were classified as “highly accurate”, “accurate”, “moderately accurate”, and “uninformative tests”, respectively. Differences in the mean values were considered significant at a 95% confidence interval (*p* ≤ 0.05).

## 5. Conclusions

This study provides the first comprehensive quantitative analysis of sulfated N-glycans in the serum of patients with BC, identifying seven potential biomarkers with high diagnostic accuracy (AUC ≥ 0.8) for early-stage BC (stages I and II). These findings highlight the critical role of glycomic alterations in BC and open new avenues for improving early detection strategies. By preserving sialic acid groups during the analysis, we achieved a more complete characterization of glycan structures, particularly those enriched in terminal Lewis-type epitopes, which are strongly implicated in cancer progression.

Although our study focused on an Ethiopian cohort, it establishes a robust foundation for future large-scale investigations—incorporating cross-validation, external replication, and correction for false discovery rates—across diverse populations and clinical settings, including the use of benign breast conditions as a control. These insights not only advance our understanding of cancer glycobiology but also reinforce the potential of sulfated glycan profiling as a powerful tool for the diagnosis of early-stage BC. Further research is warranted to validate these biomarkers in broader populations and to elucidate their mechanistic roles in BC pathogenesis.

## Figures and Tables

**Figure 1 ijms-26-04968-f001:**
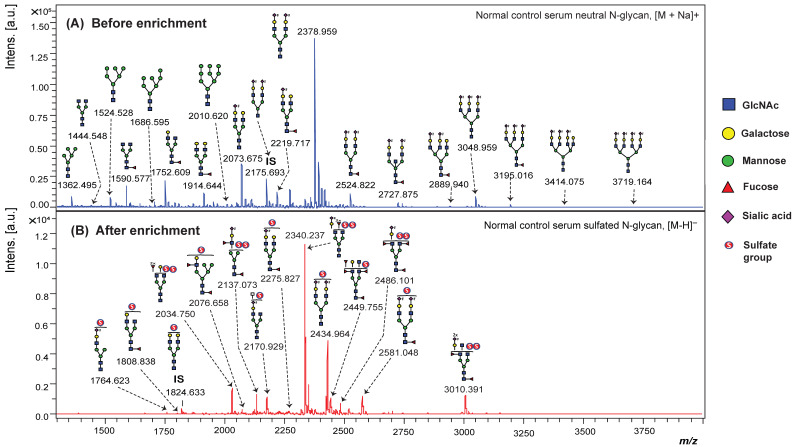
MALDI-TOF MS profiles of BOA-labeled sulfated N-glycans from healthy human serum: (**A**) before and (**B**) after enrichment using glycoblotting-based sulphoglycomics workflow. Peaks corresponding to sulfated N-glycans are highlighted. IS indicates the internal standard used for normalization. The enrichment process improves the detection of low-abundance sulfated species.

**Figure 2 ijms-26-04968-f002:**
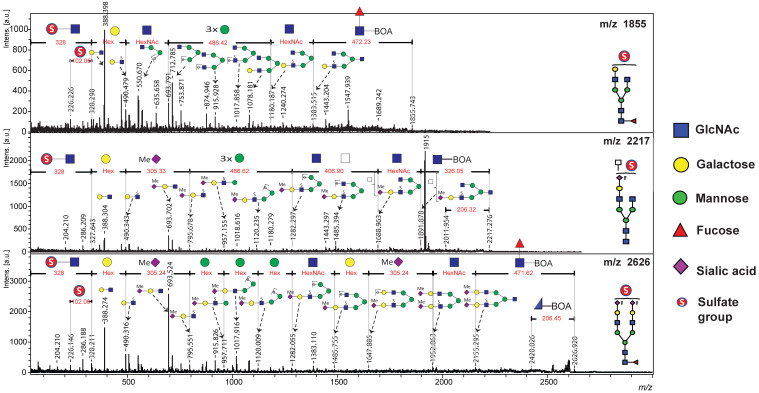
MALDI-TOF/TOF MS profiles of selected BOA-labeled sulfated N-glycans analyzed in positive-ion mode and detected as [M + 2Na − H]^+^ ions. The TOF/TOF MS spectra of mono-sulfated complex-type N-glycans at *m*/*z* 1855, 2217, and 2626 revealed their characteristic fragmentation patterns. N-glycan structures were inferred based on experimental *m*/*z* values using the Expasy GlycoMod Tool and the GlyConnect Database from the Swiss Institute of Bioinformatics. Fragment ion assignments were manually annotated using Bruker FlexAnalysis 3.0 and GlycoWorkbench.

**Figure 3 ijms-26-04968-f003:**
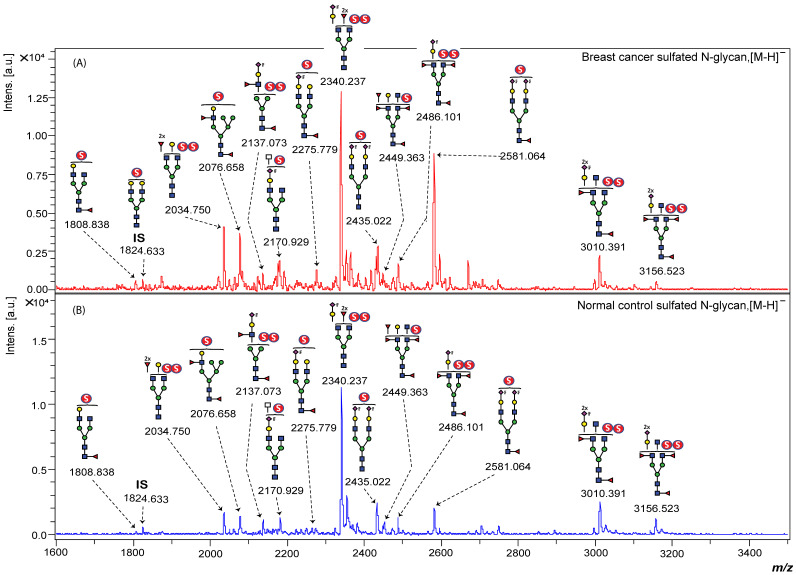
Representative MALDI-TOF MS spectra of BOA-labeled sulfated N-glycans from (**A**) a serum BC patient and (**B**) a healthy control enriched using the glycoblotting method and then fractionated using WAX. IS indicates the internal standard, mono-sulfated biantennary complex-type N-glycan (129 pmol).

**Figure 4 ijms-26-04968-f004:**
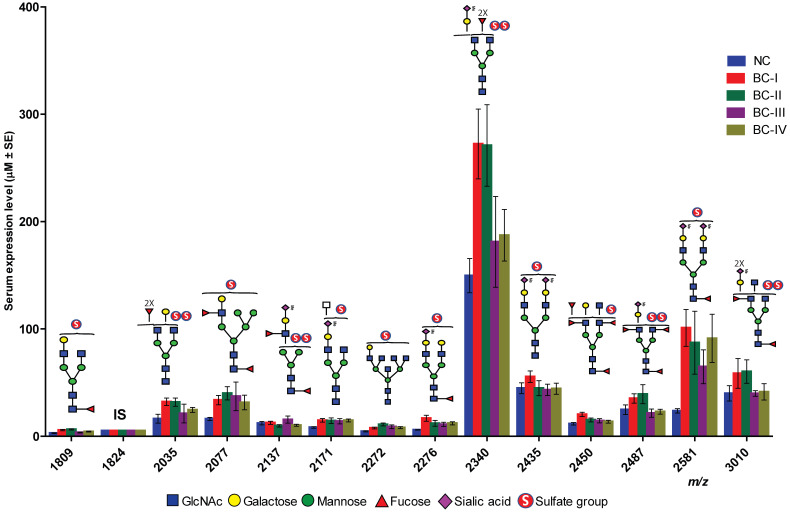
Expression levels and structures of sulfated N-glycans isolated from NC and BC serum glycoproteins. I.S. indicates an internal standard, mono-sulfated biantennary complex-type N-glycan (129 pmol).

**Figure 5 ijms-26-04968-f005:**
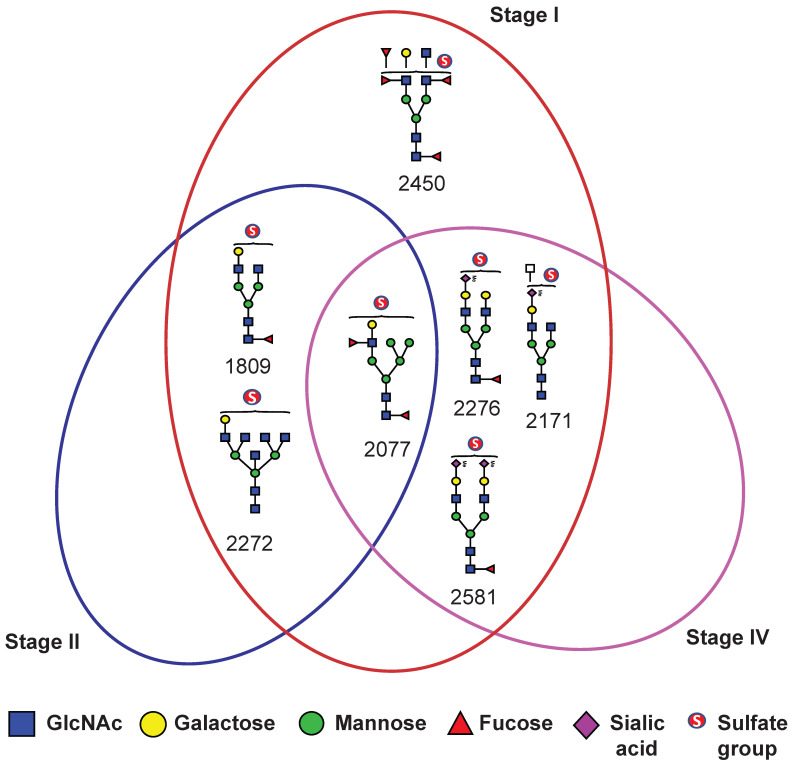
Venn diagram illustrating BC stage-specific and overlapping sulfated N-glycans identified as candidate biomarkers.

**Figure 6 ijms-26-04968-f006:**
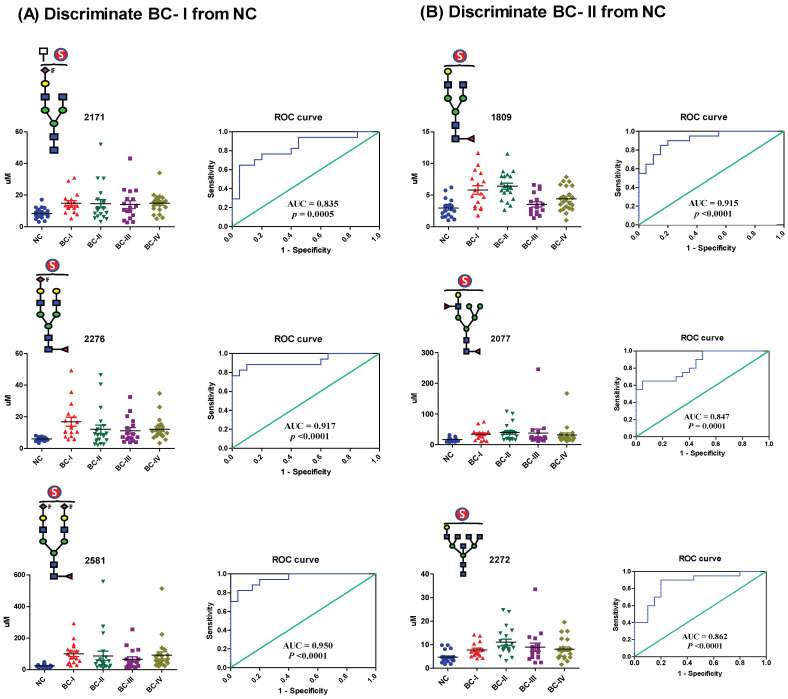
The dot plot illustrates the expression levels of serum sulfated N-glycans that were upregulated in BC patients compared to healthy controls, whereas the ROC curve displays the discriminatory power of each glycan in distinguishing between patients and NCs. The area under the curve (AUC) value indicates the effectiveness of a glycan as a diagnostic marker: (**A**) discrimination of BC-I from NC; (**B**) discrimination of BC-II from NC.

**Figure 7 ijms-26-04968-f007:**
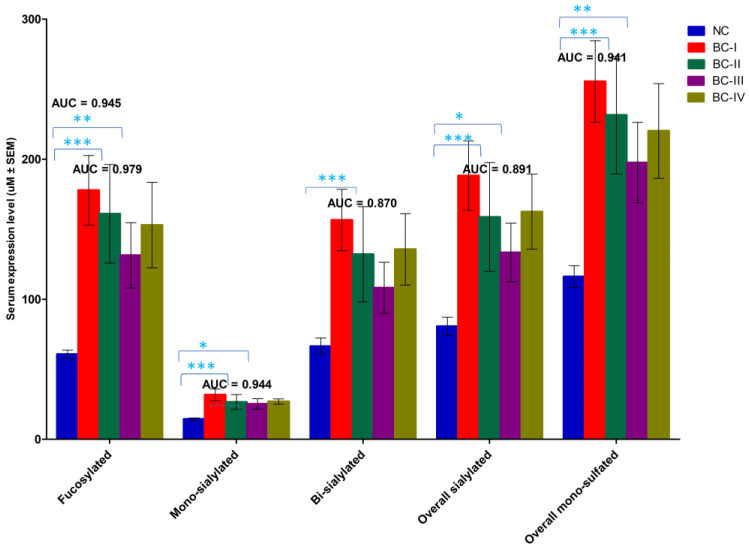
The serum glycan expression patterns of glyco-subclasses shared similar structures between the BC stages and the NC group. The area under the curve (AUC) was used to indicate the effectiveness of glycan subclasses as diagnostic markers for early-stage BC. Data are shown as mean ± SEM, * 0.01 < *p* ≤ 0.05, ** 0.0001 < *p* ≤ 0.01, *** *p* ≤ 0.0001.

**Table 1 ijms-26-04968-t001:** List of upregulated sulfated N-glycans in the serum of patients with breast cancer compared to healthy normal controls.

Serum Expression Level (AVG ± SEM)
*m*/*z*	Glycan Structure	NC	Whole BC	*p*-Value (*t*-Test)
1809	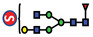	2.94 ± 0.317	5.05 ± 0.275	0.0003
2035	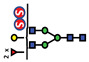	16.37 ± 1.132	27.25 ± 2.440	0.0259
2077	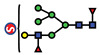	16.22 ± 1.458	35.58 ± 3.971	0.0149
2171	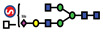	8.28 ± 0.771	14.58 ± 1.004	0.0022
2272	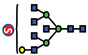	4.63 ± 0.548	8.98 ± 0.634	0.0009
2276	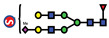	6.03 ± 0.255	12.94 ± 1.138	0.0026
2340	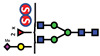	149.60 ± 15.920	227.0 ± 17.430	0.0298
2450	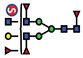	11.67 ± 1.154	15.79 ± 0.908	0.0302
2581	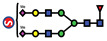	23.74 ± 2.067	86.45 ± 11.220	0.0053

All peaks corresponding to sulfated glycans exhibited a notable increase in abundance (*p* ≤ 0.05) in the sera of patients with BC compared to the NC group.

**Table 2 ijms-26-04968-t002:** Candidate sulfated N-glycan markers for different BC stages based on ROC curve analysis.

*m*/*z*	Breast Cancer vs. Normal Control										Possible Biomarker for
	BC-I (n = 17)		BC-II (n = 20)		BC-III (n = 17)		BC-IV (n = 22)		Whole BC (n = 76)		
	AUC	*p*	AUC	*p*	AUC	*p*	AUC	*p*	AUC	*p*	
1809	**0.832**	**0.001**	**0.918**	**˂0.0001**	0.597	0.314	0.745	0.006	0.772	0.0001	I, II
2077	**0.832**	**0.001**	**0.857**	**0.0001**	0.741	0.012	**0.809**	**0.0006**	**0.810**	**˂0.0001**	I, II, IV, whole BC
2170	**0.896**	**0.0001**	0.726	0.015	0.685	0.054	**0.838**	**0.0001**	0.784	0.0001	I, IV
2272	**0.803**	**0.003**	**0.855**	**0.0001**	0.744	0.011	0.750	0.005	0.786	˂0.0001	I, II
2276	**0.90**	**˂0.0001**	0.728	0.014	0.697	0.041	**0.914**	**˂0.0001**	**0.810**	**˂0.0001**	I, IV, whole BC
2450	**0.803**	**0.0029**	0.655	0.097	0.600	0.300	0.593	0.3019	0.652	0.0382	I
2581	**0.942**	**˂0.0001**	0.792	0.0018	0.708	0.030	**0.918**	**˂0.0001**	**0.840**	**˂0.0001**	I, IV, whole BC

An AUC value ≥ 0.8 and a level of significance (*p* ≤ 0.05, 95% CI) were considered the criteria for candidate biomarkers (AUC vs. diagnostic accuracy: 0.9–1 = highly accurate, 0.8–0.9 = accurate, 0.7–0.8 = moderately accurate, ˂0.7 = uninformative test). The bold AUC and *p*-values met the criteria and were selected as candidate biomarkers for their respective stages.

## Data Availability

All data are available in the Appendix A at GlycoPOST [51] (Accession ID: GPST000456) and the preprint server of medRxiv [52].

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
