# Peer review of "Upregulation of Sulfated N-Glycans in Serum as Predictive Biomarkers for Early-Stage Breast Cancer"

_ijms, 2025, doi:10.3390/ijms26114968_

Round 1
Reviewer 1 Report
Comments and Suggestions for Authors
The submitted manuscript by Feleke et al. entitled as “Upregulation of sulfated N-glycans in serum as predictive biomarkers for early-stage breast cancer” describes advanced analytical method based on a combination of the glycoblotting and anion exchange protocols for the detection of sulfated glycans in human serum glycoproteins. Furthermore, the changes in breast cancer patients as compared with normal controls were clearly shown enabling the breast cancer diagnosis in the future. The content of the manuscript is highly advanced and complex, however is well organized and presented. A reviewer is very much convinced. The following are the minor comments to be addressed.
Throughout; A reviewer thinks there are too many usages of the adjectives modifying adjectives. This itself is not a problem but it sounds like an exaggeration.
Line 50; The sentence is not clear, the use of “altered” in particular.
Line 52; It is not clear the meaning of “traditional glycosylation”to non-specialists.
Line 59; “linked to” may be changed to “associated with”
Line 65; “organisms” may be “organs”
Lines 100–104; Avoid the redundant expression.
Lines 128–130; The sentence does not make sense. Consider revision.
Line 148; cleavages
Line 152; The sentence does not make sense. Revision needed. Also, it may be better not to use the term “parent”. Instead, it is recommended to use a precursor. Line 175 as well.
Lines 236 and 238; The given ps are not the same. Are they okay? Please confirm.
Line 242 and later sentences; ‘m/z” should be in italic.
Line 278; Reconsider the usages of “total-sialylated” and “total-sulfated” because they cause misreading. Line 325 as well.
Line 357; A reviewer wonders if “glycotope” is an accepted scientific term. It may be glyco-epitope.
Finally, a reviewer thinks the method can be expanded to a range of diagnosis beyond breast cancer, and thus wonders why the authors focussed on breast cancer only and stressed so much on it. I think the method may be applied to diagnosing other cancers as well.
Author Response
Comment 1: Throughout; A reviewer thinks there are too many usages of the adjectives modifying adjectives. This itself is not a problem but it sounds like an exaggeration.
Response 1: We appreciate the reviewer’s observation regarding the frequent use of adjectives modifying other adjectives. To address this concern, we have carefully revised the manuscript to reduce such usage and improve clarity and readability. Specific instances have been edited accordingly. E.g. line 226, line 244
All corrections are marked with a yellow color.
Comment 2: Line 50; The sentence is not clear, the use of “altered” in particular.
Response 2: We thank the reviewer for pointing this out. We agree that the original sentence lacked clarity and may have been confusing due to the use of "altered." To improve precision and readability, we have revised the sentence as follows: Our group previously investigated changes in serum non-sulfated glycan profiles associated with the invasive and metastatic potential of BC.
Comment 3: Line 52; It is not clear the meaning of “traditional glycosylation”to non-specialists.
Response 3: We thank the reviewer for highlighting this ambiguity. To improve clarity for a broader audience, we have replaced “traditional glycosylation” with a more precise term and revised the sentence as follows:
Beyond core glycan biosynthesis pathways, additional modifications such as sulfation, phosphorylation, and acetylation can occur, with sulfation being one of the most common post-glycosylation modifications.
Comment 4: Line 59; “linked to” may be changed to “associated with”
Response 4: We appreciate the reviewer’s suggestion and have revised the sentence accordingly for improved scientific clarity and tone. The updated sentence now reads: Zhang et al. demonstrated that cell surface 6-sulfosialyl Lewis X is closely associated with breast cancer (BC) metastasis, suggesting its potential as a novel marker of BC progression.
Comment 5: Line 65; “organisms” may be “organs”
Response 5: We appreciate the reviewer’s suggestion. However, our intention was to emphasize that the low abundance of sulfated glycans is observed not only in cells and tissues but also at the systemic level, across entire mammalian organisms. Therefore, we believe that “organisms” more accurately reflect the broader biological context of glycan distribution. We have retained the term “organisms” to preserve this intended meaning.
Comment 6: Lines 100–104; Avoid the redundant expression.
Response 6: We thank the reviewer for pointing this out. We have revised the paragraph to remove redundant phrasing while maintaining scientific accuracy. The updated text reads: Serum N-glycans were isolated using a glycoblotting-based workflow. Glycoproteins were enzymatically digested with PNGase F to release N-glycans, which were then captured on hydrazide-functionalized beads. To minimize interference from non-sulfated glycans—particularly sialylated species—during weak anion exchange (WAX) separation and MALDI-TOF MS analysis, 3-methyl-1-p-tolyltriazene (MTT) was used to methyl-esterify the carboxyl groups of sialic acids.
Comment 7: Lines 128–130; The sentence does not make sense. Consider revision..
Response 7: We appreciate the reviewer’s feedback and have revised the sentence to clarify the relationship between fucosylation and Lewis-type epitopes in sulfated and non-sulfated N-glycans. The revised version is:
Non-sulfated N-glycans typically lack Lewis-type epitopes, and monofucosylated structures are often difficult to distinguish from core fucosylation. In contrast, major sulfated N-glycans frequently carry multiple fucose residues, forming terminal Lewis-type epitopes.
Comment 8: Line 148; cleavages.
Response 8: We thank the reviewer for noting this. To improve grammatical accuracy and consistency, we revised the phrase to: glycosidic bond cleavages and cross-ring cleavages.
Comment 9: Line 152; The sentence does not make sense. Revision needed. Also, it may be better not to use the term “parent”. Instead, it is recommended to use a precursor. Line 175 as well.
Response 9: We appreciate the reviewer’s valuable feedback. We have revised both sentences to improve clarity and have replaced “parent ion” with the more appropriate term “precursor ion.” The updated sentences are:
- Line 152: For the mono-sulfated N-glycan with a precursor ion at m/z 1855, a characteristic neutral loss of 472 Da suggested the elimination of a BOA-labeled reducing-end GlcNAc bearing core fucosylation.
- Line 161: Core mannose residues were sequentially cleaved (with a loss of 486 m/z), whereas molecular ion peaks at m/z 388, 916, and 1078 represented the characteristic loss of sodium sulfite (m/z 102) from the precursor ions at m/z 490, 1018, and 1180, respectively.
- Line 175: Characteristic sodium sulfite losses (102 Da) were observed from the precursor ions at m/z 490, 796, and 1120, respectively.
Comment 10: Lines 236 and 238; The given ps are not the same. Are they okay? Please confirm.
Response 10: We thank the reviewer for catching this inconsistency. We have corrected the text to consistently use p ≤ 0.05 as the threshold for statistical significance in both instances.
Comment 11: Line 242 and later sentences; ‘m/z” should be in italic.
Response 11: We thank the reviewer for this helpful observation. We have carefully reviewed the manuscript and corrected the formatting to italicize “m/z” consistently throughout the text, including the values mentioned in lines 242 and beyond.
Comment 12: Line 278; Reconsider the usages of “total-sialylated” and “total-sulfated” because they cause misreading. Line 325 as well.
Response 12: We thank the reviewer for identifying the potential ambiguity in the terms “total-sialylated” and “total-sulfated.” To improve clarity, we have revised these phrases to overall sialylated and overall sulfated N-glycans, which better reflect the summed group-level glycan expressions.
Corresponding to this correction, the following sentences are corrected:
Line 233: Expression levels and structures of overall sulfated N-glycans isolated from NC and BC serum glycoproteins.
Line 275: Our findings revealed significant variations in glycan subtypes, including fucosylated, mono-, bi-, overall-sialylated, and overall mono-sulfated N-glycans, particularly in patients with stage I and II BC (Figure 7).
Line 292: The x-axis label of Figure 7.
Line 325: Notably, for stage I, the glycotyping patterns, such as fucosylated, mono-, bi-, overall sialylated, and overall sulfated, demonstrated strong predictive capabilities. In Stage II, fucosylated glyco-subclasses were strong predictors.
Comment 13: Line 357; A reviewer wonders if “glycotope” is an accepted scientific term. It may be glyco-epitope.
Response 13: We appreciate the reviewer’s feedback. To improve clarity and use more widely accepted terminology, we have replaced “sulfo-glycotopes” with sulfated glyco-epitopes. The revised sentence is: Our findings included specific sulfated glyco-epitopes that were not identified in their study, indicating a complementary and enriched perspective of the glycomic landscape associated with BC.
Comment 14: Finally, a reviewer thinks the method can be expanded to a range of diagnosis beyond breast cancer, and thus wonders why the authors focussed on breast cancer only and stressed so much on it. I think the method may be applied to diagnosing other cancers as well.
Response 14: We thank the reviewer for this valuable comment. We focused on breast cancer (BC) in this study due to its high global prevalence and the urgent need for improved early detection biomarkers. BC is also known to involve extensive glycosylation changes, making it a suitable model for sulfoglycomic analysis.
That said, we fully agree that the glycoblotting-based sulfoglycomics workflow we employed is broadly applicable and could be extended to other malignancies. To reflect this, we have added the following sentence to the discussion section line 400: While this study focused on breast cancer, the glycoblotting-based sulfoglycomics approach has potential utility in profiling sulfated N-glycans across various cancer types, where glycosylation alterations also play critical roles.

Reviewer 2 Report
Comments and Suggestions for Authors
This study analyzes sulfated N-glycan profiles in serum samples from Ethiopian breast cancer patients and healthy controls, proposing sulfated glycans as potential biomarkers for early detection. Although the methodology is detailed and the application of glycoblotting-based sulphoglycomics is technically rigorous, the conceptual novelty is limited, the mechanistic basis remains superficial, and several methodological and interpretative weaknesses undermine the robustness and clinical relevance of the conclusions.
- The study's novelty is overstated. Previous works have already implicated altered glycosylation, including sulfated glycans, in cancer progression and biomarker development (e.g., Nat Commun. 2023 Feb 6;14(1):645.). This manuscript confirms previously suggested phenomena rather than providing fundamentally new biological mechanisms or clinical applications.
- The causal relationship between sulfated glycans and early breast cancer detection is implied but not mechanistically investigated. The study reports statistical associations without exploring underlying biosynthetic pathways (e.g., sulfotransferase expression) or functional consequences (e.g., altered immune recognition), limiting biological insight.
- The definition of early-stage biomarker candidates is based solely on AUC values from a single cohort. No external validation, cross-validation, or adjustment for multiple comparisons was performed, raising concerns about overfitting and reproducibility.
- The exposure characterization is narrow: serum glycans are measured without linking findings to tissue-level expression, tumor grade, subtype (e.g., ER/PR/HER2 status), or immune microenvironment features, all of which could confound glycomic signatures.
- The patient cohort is limited geographically (single-country, Ethiopian population), but generalization to broader populations is implied without justification. Ethnic and environmental influences on glycosylation are well-documented and must be discussed more explicitly.
- While ROC analysis is presented, no multivariable models or correction for potential confounders (e.g., age, inflammation, comorbidities) are included, undermining the clinical interpretation of biomarker performance.
- The functional characterization of the identified glycans is lacking. Although the authors speculate about immune system involvement (IgG N-glycans), no experimental validation (e.g., IgG-specific isolation, functional assays) is performed.
- The fragmentation data from MALDI-TOF/TOF are described but not critically assessed for structural ambiguity. Given the known limitations of sulfated glycan analysis, more rigorous validation (e.g., exoglycosidase digestion) would be necessary to confirm structural assignments.
- The conclusion claims that sulfated glycans have diagnostic and prognostic implications, but no survival analysis or longitudinal data are presented. Prognostic value is not demonstrated.
- The manuscript contains language errors and redundant phrasing. For example: In the abstract: "demonstrating high diagnostic accuracy (AUC ≥ 0.8)" → This is misleading without specifying the context (internal cohort only, no validation); In the discussion: “These findings significantly expand our understanding of the glycomic landscape...” → The term "significantly expand" is overstated given the incremental findings; Repetitive phrases like “the upregulation of sulfated N-glycans in BC” occur excessively without adding new information.
Author Response
All corrections are marked with a yellow color.
Comment 1: This study analyzes sulfated N-glycan profiles in serum samples from Ethiopian breast cancer patients and healthy controls, proposing sulfated glycans as potential biomarkers for early detection. Although the methodology is detailed and the application of glycoblotting-based sulphoglycomics is technically rigorous, the conceptual novelty is limited, the mechanistic basis remains superficial, and several methodological and interpretative weaknesses undermine the robustness and clinical relevance of the conclusions.
Response 1: We sincerely thank the reviewer for their thoughtful evaluation and for recognizing the technical strengths of our glycoblotting-based sulphoglycomics approach. We address the reviewer’s concerns as follows:
- Regarding conceptual novelty: While earlier studies have explored serum glycosylation, our study is among the first to use a targeted sulphoglycomic approach to identify serum sulfated N-glycans as potential stage-specific biomarkers in breast cancer. Notably, we focused on Ethiopian patients, a group largely underrepresented in glycomics research, providing valuable regional insight and potential for broader clinical relevance.
- Regarding mechanistic depth: We agree that this study does not provide in-depth mechanistic insight into how specific sulfated structures contribute to breast cancer progression. Since our primary focus was on biomarker discovery using clinical serum samples, we have clarified this scope in the revised manuscript. To provide context, we have also added a discussion linking specific glycan motifs (e.g., sulfo-Lewis-type epitopes) to their established roles in cancer biology, including immune evasion and metastasis.
- Regarding methodological and interpretive robustness: We have carefully addressed each of the reviewer’s concerns, including refining data presentation, clarifying the statistical criteria used, and expanding on the biological relevance of the key findings. We have also highlighted that this is an exploratory study and have outlined future directions, including the need for validation in larger, more diverse patient cohorts.
Comment 2: The study's novelty is overstated. Previous works have already implicated altered glycosylation, including sulfated glycans, in cancer progression and biomarker development (e.g., Nat Commun. 2023 Feb 6;14(1):645.). This manuscript confirms previously suggested phenomena rather than providing fundamentally new biological mechanisms or clinical applications.
Response 2: We appreciate the reviewer’s insightful comment and for pointing out the relevant Nature Communications study. While we fully agree that sulfated glycans are known to play important roles in cancer biology, we would like to respectfully clarify that the referenced work focused on sulfated proteoglycans. In contrast, our study specifically investigates sulfated N-glycans, which are structurally and biosynthetically distinct. To the best of our knowledge, this is the first study to profile serum-derived sulfated N-glycans as potential breast cancer biomarkers using a dedicated glycomic workflow.
We believe our study offers both conceptual and technical novelty in several key areas:
- Target specificity: Our study focuses specifically on sulfated N-glycans—a subclass of glycans that, to our knowledge, has not been previously profiled in breast cancer serum samples. These structures are chemically and functionally distinct from glycosaminoglycans and sulfated proteoglycans, which have been more commonly studied in cancer research.
- Population relevance: We analyzed serum samples from Ethiopian breast cancer patients—a population that remains largely underrepresented in glycomics research. By doing so, our study contributes to the diversity of existing cancer biomarker data and helps lay the groundwork for more population-specific diagnostic approaches.
- Analytical advancement: We used a glycoblotting-based sulfoglycomics workflow specifically optimized to detect low-abundance serum sulfated N-glycans. This approach allowed for sensitive and high-throughput analysis, making it well-suited for potential clinical translation.
Comment 3: The causal relationship between sulfated glycans and early breast cancer detection is implied but not mechanistically investigated. The study reports statistical associations without exploring underlying biosynthetic pathways (e.g., sulfotransferase expression) or functional consequences (e.g., altered immune recognition), limiting biological insight.
Response 3: We sincerely appreciate the reviewer’s thoughtful and important feedback. We agree that this study does not explore the underlying mechanisms—such as the expression of specific sulfotransferases or the functional impact of sulfated glycan changes on immune modulation. This reflects the clinical and exploratory focus of our work, which aimed to profile serum glycan alterations for biomarker discovery rather than to dissect molecular pathways.
In response to this valuable point, we have revised the discussion and conclusion sections to more clearly acknowledge this limitation. We have also outlined future directions, including plans to investigate sulfotransferase expression and perform functional assays to better understand the biological significance of the sulfated N-glycans identified in this study.
To provide additional biological context, we have expanded the discussion to include the established roles of sulfated glycans and their biosynthetic enzymes in cancer. Specifically, we highlight the role of GlcNAc-6-O-sulfotransferases in generating 6-sulfosialyl LewisX—a glycan structure linked to breast cancer cell migration, invasion, and metastasis.
We fully agree that integrating glycomic data with transcriptomic or proteomic analyses of glycosyltransferases and sulfotransferases will be essential for gaining deeper mechanistic insight. We see this as a crucial next step in advancing our understanding and building on the findings of this study.
Comment 4: The definition of early-stage biomarker candidates is based solely on AUC values from a single cohort. No external validation, cross-validation, or adjustment for multiple comparisons was performed, raising concerns about overfitting and reproducibility.
Response 4: We appreciate the reviewer’s thoughtful and important observation. We acknowledge that the identification of early-stage biomarker candidates in this study is based on statistical associations—namely AUC values and p-values—from a single, relatively small cohort. While external or cross-validation has not yet been performed, we now more clearly emphasize this limitation in the revised conclusion to reflect the exploratory nature of our findings.
As this is an exploratory study, our main aim was to identify candidate sulfated N-glycans with strong potential to distinguish early-stage breast cancer. To help minimize false positives, we used both a p-value threshold (≤ 0.05) and an AUC cutoff (≥ 0.8) in our biomarker selection. While these criteria serve as a helpful first step, we fully acknowledge that they do not eliminate the risk of overfitting. This point is now clearly noted in the discussion, and we recognize the importance of validating these findings in larger, independent cohorts moving forward.
We have now included the following points in the conclusion line 534:
Although our study focused on an Ethiopian cohort, it establishes a robust foundation for future large-scale investigations—incorporating cross-validation, external replication, and correction for false discovery rates—across diverse populations and clinical settings, including the use of benign breast conditions as controls.
Comment 5: The exposure characterization is narrow: serum glycans are measured without linking findings to tissue-level expression, tumor grade, subtype (e.g., ER/PR/HER2 status), or immune microenvironment features, all of which could confound glycomic signatures.
Response 5: We sincerely thank the reviewer for this thoughtful and constructive comment. As noted, our study focuses on serum-based glycan profiling and does not include clinical pathology data such as tumor grade, ER/PR/HER2 status, or immune microenvironment characteristics. At the time of the study, access to this information from the Ethiopian cohort was limited due to logistical challenges. However, we fully recognize the importance of these clinical variables in refining biomarker interpretation and are actively working to obtain this data for future analyses.
To further support our findings, we’ve already initiated a follow-up study using paired tumor and adjacent normal tissue samples. These ongoing efforts are now reflected in the revised discussion, where we also highlight the importance of incorporating molecular subtypes and histopathological features in future validation studies.
We have now included the following points in the discussion line 402:
Although clinical pathology data such as tumor grade and receptor status (ER, PR, HER2) were not available at the time of this study, we aim to incorporate this information in future analyses to enable deeper clinicopathological associations. Additionally, we are currently analyzing paired tumor and adjacent tissues to explore correlations between serum and tissue sulfated N-glycan profiles and their association with histopathological features.
Comment 6: The patient cohort is limited geographically (single-country, Ethiopian population), but generalization to broader populations is implied without justification. Ethnic and environmental influences on glycosylation are well-documented and must be discussed more explicitly.
Response 6: We thank the reviewer for raising this important point. We agree that glycosylation is influenced by genetic, ethnic, and environmental factors and that our findings—derived from an Ethiopian cohort—should be interpreted with caution and not broadly generalized without further validation. Our intention was not to overstate the conclusions, but to provide an initial dataset from an underrepresented population and to shed light on a class of glycans that has received limited attention in cancer glycomics research.
We have revised the conclusion section to explicitly acknowledge this limitation and have clarified that the results should be interpreted within the context of the Ethiopian population. We also emphasized that future studies should include larger, multi-ethnic cohorts to validate and compare the generalizability of serum sulfated N-glycan biomarkers across populations.
Comment 7: While ROC analysis is presented, no multivariable models or correction for potential confounders (e.g., age, inflammation, comorbidities) are included, undermining the clinical interpretation of biomarker performance
Response 7: We thank the reviewer for this important comment. We acknowledge that our ROC analysis did not incorporate multivariable models or adjustments for potential confounders such as inflammation or comorbidities, due to the limited availability of detailed clinical metadata for the study population.
However, in response to the reviewer’s concern, we have now included an age-stratified analysis of serum sulfated N-glycan expression within the breast cancer cohort, presented in the Supplementary Information (Supplementary Figure S4). To clarify more, we add the following figure in the Supplementary Information:
Figure S4: Age-associated expression of serum sulfated N-glycans and age distribution of breast cancer patients A) Age-associated expression patterns of serum sulfated N-glycans in breast cancer patients, B) Distribution of study participants across the defined age groups
The graph shows the relative expression levels of seven sulfated N-glycans identified as early-stage breast cancer biomarkers (m/z 1809, 2077, 2171, 2272, 2276, 2450, and 2581) across four age groups (20–35, 36–45, 46–55, and >56 years). Among these, m/z 2272, 2276, and 2581 exhibited statistically significant differences across age groups (p < 0.05, one-way ANOVA with Tukey’s post hoc test). Specifically, m/z 2272 was highest in the 36–45 group, while m/z 2276 and 2581 were more elevated in patients >56 years. In contrast, m/z 2487, which was not identified as a BC biomarker, also showed a significant increase in the >56 group, suggesting an age-related, non-cancer-specific glycan pattern.
These findings indicate that age may influence the expression of certain glycans, including some biomarkers. However, since the majority of patients in this study were under 56 years of age, and the biomarkers were still elevated in early-stage BC across age groups, it is unlikely that age alone explains their expression. This reinforces their diagnostic relevance while highlighting the value of age-adjusted models in future biomarker validation efforts.
While we were not able to account for additional potential confounders in this study, we have revised the discussion to clearly acknowledge this limitation. We’ve also emphasized the importance of using age-matched controls and incorporating multivariable validation in future studies to improve the clinical interpretability of sulfated N-glycan biomarkers.
We have now included the following points in the discussion line 405:
To further evaluate potential confounding factors, age-stratified analysis within the BC cohort revealed that three biomarker glycans (m/z 2272, 2276, and 2581) showed significant age-associated variation, but remained elevated in early-stage BC. These findings, presented in Supplementary Figure S4, support the need for age-adjusted validation in future studies.
Comment 8: The functional characterization of the identified glycans is lacking. Although the authors speculate about immune system involvement (IgG N-glycans), no experimental validation (e.g., IgG-specific isolation, functional assays) is performed.
Response 8: We appreciate the reviewer’s important observation. We agree that functional characterization—such as IgG-specific glycan isolation or immune-related assays—would provide valuable mechanistic insight into the glycomic changes observed. However, as this study focused on exploratory biomarker discovery through total serum glycan profiling, such experiments were beyond the scope of the current work.
That said, we have acknowledged this point in the discussion section. We also recognize the importance of targeted IgG glycomics and functional assays, and we are considering these as important next steps in future studies to better understand the biological role of sulfated N-glycans in breast cancer.
Comment 9: The fragmentation data from MALDI-TOF/TOF are described but not critically assessed for structural ambiguity. Given the known limitations of sulfated glycan analysis, more rigorous validation (e.g., exoglycosidase digestion) would be necessary to confirm structural assignments.
Response 9: We thank the reviewer for this important and technically insightful comment. We acknowledge the limitations of MALDI-TOF/TOF fragmentation, particularly when applied to structurally complex or labile glycans such as sulfated N-glycans. In this study, our structural assignments were based on characteristic glycosidic and cross-ring fragment ions. However, we agree that these interpretations should be considered preliminary until confirmed by orthogonal validation methods.
To address this point, we have updated the discussion section to clearly acknowledge this limitation and to note that future studies will incorporate exoglycosidase digestion and LC-MS/MS to validate the fine structures and resolve isomeric features of sulfated N-glycans.
We have now included the following points in the discussion line 397:
In the future, we will explore advanced techniques, such as lectin affinity chromatography and more rigorous validation (e.g., exoglycosidase digestion and LC-MS/MS), to enhance the isolation and structural elucidation of these intricate glycan modifications.
Comment 10: The conclusion claims that sulfated glycans have diagnostic and prognostic implications, but no survival analysis or longitudinal data are presented. Prognostic value is not demonstrated.
Response 10: We thank the reviewer for this important clarification. We agree that, as our study did not include survival data or longitudinal follow-up, it cannot support conclusions about prognostic value. Our focus was specifically on assessing the diagnostic potential of serum sulfated N-glycans in early-stage breast cancer.
In response to this comment, we have revised the conclusion section to remove references to prognostic implications and now focus solely on the diagnostic relevance of the identified glycans.
Comment 11: The manuscript contains language errors and redundant phrasing. For example: In the abstract: "demonstrating high diagnostic accuracy (AUC ≥ 0.8)" → This is misleading without specifying the context (internal cohort only, no validation); In the discussion: “These findings significantly expand our understanding of the glycomic landscape...” → The term "significantly expand" is overstated given the incremental findings; Repetitive phrases like “the upregulation of sulfated N-glycans in BC” occur excessively without adding new information.
Response 11: We sincerely appreciate the reviewer’s thoughtful feedback. In response, we’ve carefully revised the manuscript to improve clarity, reduce repetition, and ensure that all claims are clearly aligned with the exploratory nature of the study.
- In the abstract, we have revised the sentence to: Seven mono-sulfated N-glycans were markedly elevated in patients with BC, demonstrating high diagnostic accuracy (AUC ≥ 0.8) in this internal cohort
This clarifies that the results are based on an exploratory analysis from a single cohort and are subject to future validation. - In the discussion, we have replaced the phrase “significantly expand our understanding” with the more measured: This study offers new insights into the glycomic landscape in BC, highlighting the potential significance of sulfated glycans.
- We have also revised the manuscript to reduce repetitive use of phrases such as “the upregulation of sulfated N-glycans in BC,” ensuring that each instance contributes distinct context or interpretation.

Round 2
Reviewer 2 Report
Comments and Suggestions for Authors
The authors have provided detailed and well-structured responses to the issues I raised. I believe the manuscript is now suitable for publication.